# Effects of Hydroxysafflor Yellow A (HSYA) on UVA-Induced Damage in HaCaT Keratinocytes

**DOI:** 10.3390/ijms25147573

**Published:** 2024-07-10

**Authors:** Szu-Chieh Yu, Wan-Chun Chiu, Pei Yu Loe, Yi-Wen Chien

**Affiliations:** 1Department of Nutrition and Health Sciences, Taipei Medical University, Taipei 11031, Taiwan; angela1018@hotmail.com.tw (S.-C.Y.); wanchun@tmu.edu.tw (W.-C.C.); ma07111019@tmu.edu.tw (P.Y.L.); 2Research Center of Geriatric Nutrition, College of Nutrition, Taipei Medical University, Taipei 11031, Taiwan; 3Department of Nutrition, Wan Fang Hospital, Taipei Medical University, Taipei 11696, Taiwan; 4Graduate Institute of Metabolism and Obesity Sciences, Taipei Medical University, Taipei 11031, Taiwan; 5Nutrition Research Center, Taipei Medical University Hospital, Taipei 11031, Taiwan; 6TMU Research Center for Digestive Medicine, Taipei Medical University, Taipei 11031, Taiwan

**Keywords:** UVA, hydroxysafflor yellow A (HSYA), oxidative stress, HaCaT keratinocytes, MMPs, COX-2

## Abstract

To assess the effects of hydroxysafflor yellow A (HSYA) on ultraviolet A (UVA)-induced damage in HaCaT keratinocytes. HaCaT keratinocytes were UVA-irradiated, and the effects of HSYA on cell viability, reactive oxygen species (ROS) generation, lipid peroxidation, and messenger (m)RNA expression were measured. mRNA expressions of matrix metalloproteinase (MMP)-1, MMP-2, MMP-9, and cyclooxygenase (COX)-2 were determined by a real-time polymerase chain reaction (RT-PCR). UVA exposure led to a decrease in cell viability and an increase in ROS generation in HaCaT keratinocytes. HSYA effectively increased the viability of HaCaT keratinocytes after UVA exposure and protected them from UVA-induced oxidative stress. Moreover, HSYA inhibited expressions of MMP-1, MMP-2, MMP-9, and COX-2 by HaCaT keratinocytes with UVA-induced photodamage. Our results suggest that HSYA can act as a free radical scavenger when keratinocytes are photodamaged. HSYA has the potential to be a skin-protective ingredient against UVA-induced photodamage.

## 1. Introduction

Ultraviolet (UV) radiation from sunlight causes several types of acute and chronic skin damage, which can result in inflammation, immune changes, physical changes, and DNA damage that facilitate skin aging and the development of skin carcinogenesis [1]. UV radiation is composed of UVA, ultraviolet B (UVB), and ultraviolet C (UVC) components based on the wavelength, with UVA having the longest wavelengths (320~400 nm), UVB being mid-range (290~320 nm), and UVC having the shortest wavelengths (100~290 nm) [2]. UVA is responsible for numerous biological effects on the skin, including wrinkles, a leathery texture, and pigmentation [3]. Reactive oxygen species (ROS) are generated by excessive solar UV radiation, resulting in oxidative damage to cellular components, proteins, lipids, and nucleic acids, which accelerate aging. Therefore, it will increase the risk of skin cancer [4]. Thus, antioxidants play an important role in protecting skin against ROS-induced injury. Numerous antioxidants, such as vitamin C, vitamin E, and β-carotene, have been incorporated into many skin care products [5,6]. In recent years, natural compounds have gained considerable attention as skin protective agents [7].

Traditional Chinese medicine is gradually garnering increased attention because of its beneficial features, including numerous target effects, fewer side effects, and greater safety. So, it is necessary to explore ingredients from traditional Chinese medicines using modern technology. Carthamus tinctorius L. is commonly known as safflower and is an important traditional medicine containing abundant flavones. Hydroxysafflor yellow A (HSYA) is the main active ingredient in safflower [8], is a part of quinochalcone, and has unique structures of hydroxyl groups that provide an antioxidant effect. The multiple hydroxyl groups in its molecule may provide a source of antioxidant capacity, thus inhibiting the rupture of red blood cells induced by hydroxyl free radicals. The clinical research of HSYA mainly focuses on cerebrovascular diseases. It mainly inhibits platelet aggregation by competitively inhibiting the binding of platelet activating factors and platelet receptors to reduce the scope of brain embolism and damage to brain neurons. Previous studies proved that safflower has great effects on ameliorating oxidative stress (an imbalance between the production of ROS and the body’s ability to detoxify these reactive intermediates or repair the resulting damage), exhibited antioxidant [9] and anti-inflammatory properties [10], and promoted peripheral flow and inhibited platelet aggregation [11]. The antioxidant activity of polyphenols, including HSYA, is well-documented. Polyphenolic compounds are known for their ability to donate hydrogen atoms or electrons, thereby neutralizing ROS and protecting cellular components from oxidative damage. This protective effect is particularly relevant in the context of skin photoaging, where oxidative stress plays a critical role.

In summary, the study focuses on the effects of UVA-induced photoaging in keratinocytes, utilizing HSYA as a protective agent due to its potent antioxidant properties derived from its polyphenolic structure. However, the effects of HSYA on the skin remain unclear. Therefore, this study investigated the effects of HSYA on skincare.

## 2. Results

### 2.1. HSYA Exhibited No Cytotoxicity toward HaCaT Keratinocytes

Before the experiment, a cell viability assay was used to evaluate the toxic effect of HSYA on HaCaT keratinocytes. Our laboratory established photoaging UVA-induced skin-cell-damage cell-mode-analysis photoaging indicators, including the determination of the content of intracellular reactive oxygen species and antioxidant enzymes, and the keratinocyte-related message delivery path. In this study, we used human keratinocyte photodamage and UVA-induced stimulation of the human melanoma cell activation mode to assess the photo-aging property of safflower yellow pigment. We cultured human dermal Hs68 cells, which were human foreskin fibroblasts before differentiation. In this project, we focused on HaCaT keratinocytes. Figure 1A shows that no significant change in the cell viability of HaCaT keratinocytes occurs after treatment with HSYA at concentrations of 1~200 μM. This result indicates that HSYA is safe for HaCaT keratinocytes at the concentrations used in this study.

### 2.2. HSYA Increased HaCaT Keratinocyte Viability after UVA Exposure

The cell viability assay after UVA exposure showed that the viability of HaCaT keratinocytes significantly increased (*p* < 0.05) by pretreating cells with HSYA at concentrations of 50, 100, and 200 μM (Figure 1B). The cytotoxicity after the UVA exposure of HaCaT keratinocytes pretreated with HSYA was significantly lower (*p* < 0.05) than the control (without treatment) (Figure 2). These observations indicate that HSYA is effective in preventing UVA-induced cell damage.

### 2.3. HSYA Reduced UVA-Induced Intracellular ROS Production in HaCaT Keratinocytes

The amount of intracellular ROS in HaCaT keratinocytes was measured using 2′,7′-Dichlorofluorescein diacetate (DCFH_2_-DA), a dye that is known to react with H_2_O_2_ in the presence of peroxidase and is used to detect intracellular ROS. The results indicate that intracellular ROS production significantly increases in UVA-exposed cells. Treatment with 50, 100, and 200 μM of HSYA significantly inhibited intracellular ROS production compared to UVA-exposed cells without HSYA treatment (Figure 3A, *p* < 0.05), which indicates that HSYA can prevent intracellular ROS production after HaCaT keratinocytes are irradiated with UVA.

### 2.4. HSYA Inhibited UVA-Induced Lipid Peroxidation (LPO) in HaCaT Keratinocytes

The results indicate that LPO production significantly increases in UVA-exposed cells. Treatment with HSYA significantly inhibited intracellular LPO production compared to UVA-exposed cells without HSYA treatment (Figure 3B, *p* < 0.05), which indicates that HSYA can prevent intracellular LPO production after HaCaT keratinocytes are irradiated with UVA.

### 2.5. Effects on Expressions of MMP-1, MMP-2, MMP-9, and COX-2

As the results show, MMP-1, MMP-2, and MMP-9 genes increased after UVA irradiation in HaCaT keratinocytes. However, the expressions were reduced by treatment with HSYA (Figure 4). The results show that HSYA can inhibit UVA-induced expressions of MMP-1, MMP-2, and MMP-9 in HaCaT keratinocytes.

UVA radiation triggered an increase in COX-2 expression in HaCaT keratinocytes, and treatment with HSYA reduced that increase (Figure 5). The reduction in COX-2 expression suggests that HSYA may suppress the inflammatory process and protect against the damaging effects of UVA irradiation.

## 3. Discussion

UV irradiation, a well-known culprit of skin damage, triggers a cascade of events leading to cellular harm. Ultraviolet A (UVA) radiation is known to induce oxidative stress, leading to cellular damage through the generation of reactive oxygen species (ROS). Our study aimed to explore the protective effects of HSYA on UVA-induced damage in HaCaT keratinocytes. The results demonstrate that HSYA can mitigate some of the harmful effects caused by UVA exposure. Upon exposure to UV rays, the skin undergoes a process where ROS are generated. These ROS molecules induce lipid peroxidation in the intricate phospho-lipid structure of cell membranes, compromising their integrity [12]. LPO, in essence, acts as a saboteur to the very integrity and functionality of cell membranes. Its increased presence disrupts cell fluidity, inactivates membrane enzymes, elevates ion permeability, and can ultimately lead to the rupture of these cellular barriers. [13]. Hence, the importance of antioxidants in preventing such cellular damage cannot be overstated [14,15]. A notable study [16] shed light on the potential of HSYA in combatting cellular damage. This study revealed that HSYA exhibited remarkable efficacy in enhancing the activity of key antioxidant enzymes, like superoxide dismutase (SOD) and glutathione peroxidase (GPx). Furthermore, it was found to substantially diminish the levels of ROS and LPO in cases of alcohol-induced liver injury in rats. Building upon this groundwork, our study uncovered the protective effects of HSYA against UVA irradiation. Notably, we observed a significant reduction in the production of ROS and the generation of LPO upon HSYA intervention. Consequently, the inhibition of ROS and LPO by HSYA serves as a pivotal strategy in mitigating the risk factors associated with UVA-induced skin damage. UVA exposure can be harmful, but HSYA treatment mitigates such effects.

Kong et al. marked the genesis of our understanding regarding the protective prowess of HSYA against the deleterious effects of UV irradiation-induced photoaging in mouse skin [17]. Their seminal study illuminated the intricate mechanisms underlying HSYA’s protective action, particularly its remarkable ability to scavenge free radicals. Through topical application, HSYA treatment demonstrated multifaceted benefits, including the augmentation of collagen content, suppression of epidermal hyperproliferation, and preservation of dermal collagen and elastic fibers. These effects were attributed to HSYA’s capacity to counteract photo-oxidative stress-induced skin injury by bolstering the activities of key antioxidant enzymes, such as SOD, GPx, and catalase (CAT), while concurrently diminishing the levels of MDA, a marker of lipid peroxidation. Malondialdehyde (MDA) is a byproduct of lipid peroxidation, which occurs when ROS attack polyunsaturated fatty acids in cell membranes. In a bid to comprehensively elucidate HSYA’s radical scavenging prowess, we embarked on an investigation to assess its radical scavenging ability (RSA) across various concentrations. Our findings unveiled the impressive IC50 values of HSYA for scavenging the superoxide anion and hydroxyl radical (·OH), indicative of its robust scavenging capabilities at concentrations as low as 75.18 μg and 2.52 μg, respectively. Furthermore, our exploration into HSYA’s peroxyl radical (ROO·) scavenging ability, quantified at 0.4 mM, revealed a substantial Trolox equivalent unit (TEU) of 4.6, underscoring its potent antioxidative potential. Additionally, HSYA exhibited pronounced scavenging activity against 1,1-diphenyl-2-picrylhydrazyl (DPPH) radicals and demonstrated a notable reducing power on the Fe (3+)/ferricyanide complex at a concentration of 10 μM. In summation, our findings corroborate the notion that HSYA serves as a promising candidate in the realm of natural antioxidants, boasting remarkable free radical scavenging activity in vitro. Building upon this foundation, our study delved into the effects of HSYA on UVA-induced damage, employing HaCaT keratinocytes as a model system to unravel its therapeutic potential in combating skin damage elicited by UVA radiation. Our findings align with previous studies that highlight the harmful effects of UVA on skin cells. HSYA’s ability to increase cell viability (Figure 1B) and reduce cytotoxicity (Figure 2) after UVA exposure indicates its protective role. This is further supported by the significant reduction in intracellular ROS production (Figure 3A) and lipid peroxidation (Figure 3B) upon HSYA treatment, suggesting that HSYA can act as an effective antioxidant.

Exposure of the skin to UV irradiation produces oxidative stress and triggers synthesis of MMPs [18]. The generation of MMPs enhances the degradation of extracellular matrix (ECM) proteins, thus reducing the elasticity and integrity of the skin [19]. MMP-1 is produced by dermal fibroblasts and epidermal keratinocytes, cleaves type I collagen into fragments, and is then hydrolyzed by MMP-2 and MMP-9 [20]. MMP-2 and MMP-9 are the main enzymes that degrade type Ⅳ collagen, the major component of basement membranes. They can destroy the ECM and stimulate an inflammatory response. Thus, MMP-2 and MMP-9 play important roles in inflammatory skin diseases [21]. As shown in a previous study [22], HSYA can downregulate MMP-1 expression and contributes to the regulation of vascular proliferation responses. Another study [23] found that the injection of HSYA into Sprague Dawley rats regulated gene expressions of MMP-9 and TIMP-1 to reduce liver fibrosis in chronic liver disease. In our study, we clearly found that HSYA had effects against UVA irradiation through inhibiting the expressions of MMP-1, MMP-2, and MMP-9. Our results show that HSYA can inhibit UVA-induced expressions of MMP-1, MMP-2, and MMP-9.

In terms of the molecular mechanisms, our results show that HSYA inhibits UVA-induced increases in MMP-1, MMP-2, MMP-9, and COX-2 expressions (Figure 4 and Figure 5). These enzymes are associated with extracellular matrix degradation and inflammation. The suppression of these markers by HSYA indicates its potential to protect against UVA-induced skin aging and inflammatory responses. However, it is essential to clarify that while COX-2 is involved in inflammatory processes, its role is not exclusively linked to cancer promotion. Several studies revealed that the expression of COX-2 can be induced in skin cells by UVA irradiation [24]. Excessive exposure to sunlight causes the acute inflammation of the skin, erythema, and an immunosuppressive response, and increases the risk of skin cancer [25,26]. The process of inflammation is mainly due to nitric oxide and prostaglandins, the product of the second type of COX. We thought HSYA-mediated changes in nitric oxide levels in UVA-treated HaCaT keratinocytes; however, we need detect protein expression in our future studies. Under high oxidative stress, cytokines stimulate inflammation factors causing the expression of COX-2 and converts arachidonic acid into prostaglandins, such as prostaglandin E2 (PGE2). PGE2 is the main product of COX-2, which is considered to be a cancer promoter [27]. In our study results, we found that HSYA had effects against UVA irradiation through the inhibition of the expression of COX-2. We suggest that HSYA can reduce the inflammation induced by UVA by lowering the expression of COX-2. Finally, the role of HSYA as an ROS scavenger versus its potential to enhance an Nrf2-mediated antioxidant response warrants further investigation. Our study primarily indicates that HSYA acts as a direct ROS scavenger. Still, the involvement of antioxidant enzymes suggests a more complex mechanism that may include the modulation of the cellular antioxidant response [28,29].

In summary, our study provides evidence that HSYA can protect HaCaT keratinocytes from UVA-induced oxidative damage by reducing ROS production, lipid peroxidation, and the expressions of MMPs and COX-2. Future studies should explore the precise molecular pathways through which HSYA exerts its protective effects, including its potential role in enhancing the antioxidant response.

## 4. Materials and Methods

### 4.1. Cell Culture

HaCaT keratinocytes were cultured in a humidified incubator at 37 °C and 5% CO_2_ in Dulbecco’s modified Eagle’s medium (DMEM) supplemented with 10% fetal bovine serum (FBS), 100 units/mL of penicillin, and 100 μg/mL of streptomycin.

### 4.2. Drug Treatment and UVA Irradiation

Dissolve 250 mg of MTT powder in PBS. Then, filter the solution with a 0.2 µM filter, aliquot into microcentrifuge tubes, add 10 times the MTT stock solution (5 mg/mL), and store at −20 °C.

HaCaT cells were cultured in a 96-well plate, with 2 × 10^4^ cells seeded in each well. After culturing for 24 h in a constant-temperature incubator at 37 °C with 5% CO_2_, rinse with PBS to remove residual culture medium. Then, add different concentrations containing the dosage of HSYA in maintenance medium and culture for another 24 h. After two washes with phosphate-buffered saline (PBS), the cells were then incubated with 1 mL of PBS. Cells were irradiated with UVA at 200 mJ/cm^2^ using a UV lamp (XL-1500 UV CROSSLINKER, Columbus, OH, USA). Replace the PBS with fresh medium and incubate for an additional four hours. Finally, perform the analysis.

### 4.3. Cell Viability Assay

An MTT assay was performed to determine the number of viable cells in culture. Different concentrations (0, 1, 10, 25, 50, 100, and 200 µM) of HSYA were added to keratinocytes. The cells were incubated for 24 h. The absorbance was measured at 570 nm in a plate reader to determine the formazan concentration, which is proportional to the number of living cells in culture.

### 4.4. Lactate Dehydrogenase (LDH) Assay

The LDH assay followed the procedures described in the LDH cytotoxicity assay kit (G-Biosciences cat. 786-210, 786-324). The assay plate was incubated for 24 h in a humidified chamber at 37 °C and 5% CO_2_. At 45 min before harvesting the supernatant, 10 μL of 10× lysis buffer was added to the wells containing the target cell maximum LDH release control. At the end of incubation, the plate was centrifuged at 250× *g* for 5 min. A reconstituted substrate mix at 50 μL was added to each well of the plate and mixed thoroughly. The plate was covered with foil to protect it from light and incubated at 37 °C for 20 min. Stop solution at 50 μL was added to each well and thoroughly mixed, and the absorbance was recorded at 490 nm within an hour after the addition of the stop solution. The LDH assay was performed to assess cytotoxicity by measuring the release of LDH from damaged cells into the culture medium. The assay was conducted as follows: 1. Preparation of cell cultures: HaCaT keratinocytes were cultured in 96-well plates at a density appropriate for the experiment and allowed to adhere overnight. 2. Experimental treatments: cells were exposed to different treatments, including control (untreated), UVA exposure, and UVA exposure with various concentrations of HSYA. 3. Positive control for maximum LDH release: to determine the maximum LDH release, a subset of wells containing HaCaT keratinocytes was treated with a lysis buffer provided in the LDH assay kit to induce complete cell lysis. These wells served as the positive control and represented 100% LDH release. 4. Assay procedure: after the treatment period, the cell culture medium from each well was collected and transferred to a new 96-well plate. The LDH assay reagent was added to each well containing the collected medium, according to the manufacturer’s instructions. The plate was then incubated at room temperature for the specified time to allow the colorimetric reaction to occur. 5. Measurement: the absorbance of the samples was measured using a microplate reader at the wavelength specified in the assay protocol (usually around 490 nm). The absorbance values were proportional to the amount of LDH released into the medium. 6. Calculation of cytotoxicity: the percentage of LDH release for each sample was calculated. 7. Y-axis scale clarification: the Y-axis scale in Figure 2 represents the percentage of LDH release relative to the maximum LDH release control, which is set as 100%. This allows for a direct comparison of cytotoxicity levels across different treatments. By following this detailed procedure, the LDH assay results provide a clear and quantifiable measure of cytotoxicity induced by the treatments. The positive control ensures that the maximum LDH release is accurately determined, providing a reference point for calculating the relative LDH release in experimental samples.

### 4.5. Intracellular ROS Assay

HaCaT keratinocytes were cultured in a 12-well plate at a density 2 × 10^5^ cells per well to determine ROS. After 3 h of UVA irradiation, the production of intracellular ROS was determined by a 2,7-dichlorofluorescin diacetate (DCFH2-DA) assay. After DCFH2-DA (10 μM per 1 mL of medium) treatment, the plate was covered with foil to protect it from light and incubated at 37 °C for 30 min. The DCFH2-DA was removed and washed three times with PBS. Triton-100 (500 μL at 0.25%) was added to each well for 10 min, and the absorbance was detected at an excitation wavelength of 485 nm and an emission wavelength of 528 nm. The detection of intracellular reactive oxygen species (ROS) was performed using the fluorescent probe 2′,7′-dichlorofluorescin diacetate (DCFH-DA). This assay measures the presence of ROS within cells by detecting the fluorescence of the oxidized form of DCFH-DA. 1. Preparation of cell cultures: HaCaT keratinocytes were seeded in 96-well plates and allowed to adhere overnight. 2. Loading cells with DCFH-DA: cells were incubated with 10 μM of DCFH-DA (Sigma-Aldrich, St. Louis, MO, USA) in serum-free medium for 30 min at 37 °C in the dark. DCFH-DA is a non-fluorescent compound that diffuses into cells and is deacetylated by cellular esterases to non-fluorescent 2′,7′-dichlorofluorescin (DCFH). 4. Removal of excess DCFH-DA: after incubation, the cells were washed three times with phosphate-buffered saline (PBS) to remove any excess extracellular DCFH-DA. This step ensures that only intracellular DCFH is present, which can be oxidized by ROS to form the fluorescent compound 2′,7′-dichlorofluorescin (DCF). 5. Experimental treatments: following the washing steps, the cells were subjected to experimental treatments, including control (untreated), UVA exposure, and UVA exposure with various concentrations of HSYA. 6. Measurement of ROS: after the treatment period, the fluorescence intensity of DCF was measured using a fluorescence microplate reader with excitation and emission wavelengths of 485 nm and 535 nm, respectively. The fluorescence intensity was directly proportional to the amount of intracellular ROS. 7. Calculation: the fluorescence readings from treated cells were compared to those of the control cells to determine the relative increase in ROS production. The results are expressed as a percentage of ROS levels relative to the control group [30].

### 4.6. Lipid Peroxidation (LPO) Assay

The cells were collected and suspended in cold PBS. A cell homogenate was prepared with a homogenizer. After centrifugation, the supernatant was removed, and the terminal product of LPO, malondialdehyde (MDA), was measured to estimate the extent of LPO. Levels of MDA were determined by the addition of thiobarbituric acid (TBA). The supernatant of each sample was mixed with 1/12 N H_2_SO_4_, 10% phosphotungstic acid, and 0.67% TBA reaction reagent, and then incubated at 95 °C for 70 min. After cooling down for 10 min, the supernatant was removed and measured by a microplate reader at a wavelength of 532 nm [31].

### 4.7. RNA Isolation and Quantification of Gene Expression

Total RNA was extracted from HaCaT keratinocytes using TRIzol reagent (cat. 15596018, Thermo Fisher Scientific, Waltham, MA, USA). The purity of the RNA was measured at A260 nm/A280 nm with ratio values of 1.8~2.0. Complementary (c)DNA was synthesized from 1 μg of RNA using a high-capacity RNA to cDNA reverse transcriptase kit at a total 12 μL reaction volume. Reverse transcription was performed with sample incubation at 42 °C for 1 h, followed by 70 °C for 5 min and 4 °C for another 5 min. All polymerase chain reactions (PCRs) were performed using a thermal cycler (7300 Applied Biosystems). The ΔΔCT method was used to measure relative quantification, where values were normalized to the reference gene (GAPDH).

(1)COX-2Forward primer: TTCAAATGAGATTGTGGGAAAATReverse primer: AGATCATCTCTGCCTGAGTATCTT

(2)MMP-1Forward primer: CAGAGATGAAGTCCGGTTTTTCReverse primer: GGGGTATCCGTGTAGCACAT

(3)MMP-2Forward primer: GGCCAAGTGGTCCGTGTGReverse primer: GAGGCCCCATAGAGCTCC

(4)MMP-9Forward primer: CACTGTCCACCCCTCAGAGCReverse primer: GCCACTTGTCGGCGATAAGG

### 4.8. Statistical Analysis

All results are expressed as the mean ± standard deviation (SD). The statistical significance of differences among groups was determined by a one-way analysis of variance (ANOVA) and *t*-test using the SPSS package program ver. 19.0. Results were considered significant if the value of *p* was <0.05, according to Tukey’s post-hoc test.

## 5. Conclusions

Our study demonstrated the protective effects of HSYA in HaCaT keratinocytes against UVA-induced damage by suppressing the UVA-induced production of ROS and LPO and expressions of MMP-1, MMP-2, MMP-9, and COX-2. These suppressive abilities may have contributed to the increased cell viability after UVA irradiation. Our results suggest that HSYA can act as a free radical scavenger when keratinocytes are photodamaged. HSYA may have potential as a skin protective ingredient against UVA-induced photodamage.

## Figures and Tables

**Figure 1 ijms-25-07573-f001:**
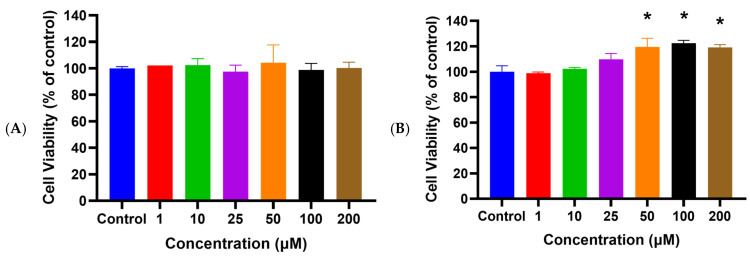
Cell viability of HaCaT cells after being treated with different concentrations of HSYA for 24 h (**A**) without UVA irradiation (**B**) were exposed to UVA (200 mJ/cm^2^) irradiation by using the MTT assay. Values are expressed as mean ± SD (*n* = 3). Values are not significantly different at * *p* < 0.05, determined by one-way ANOVA followed by Tukey’s post hoc test.

**Figure 2 ijms-25-07573-f002:**
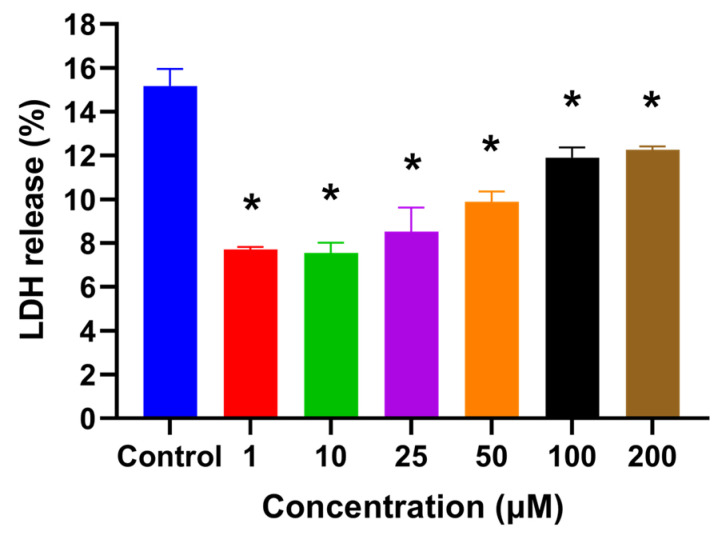
Cell cytotoxicity of HaCaT cells after being treated with different concentrations of HSYA were exposed to UVA (200 mJ/cm^2^) irradiation for 24 h by using the LDH assay. Values are expressed as mean ± SD (*n* = 3). Values with the symbol * are significantly different at *p* < 0.05, determined by one-way ANOVA followed by Tukey’s post hoc test.

**Figure 3 ijms-25-07573-f003:**
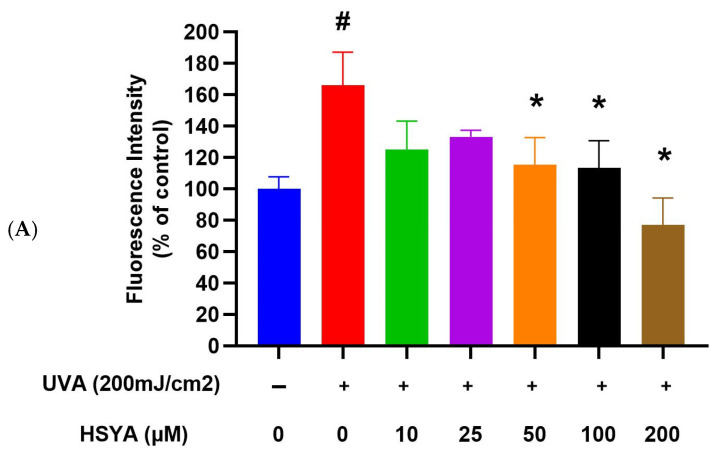
Effects of UVA-induced intracellular oxidative stress after being treated different concentrations of HSYA. (**A**) ROS generation and (**B**) lipid peroxidation generation in HaCaT cells after being treated different concentrations of HSYA exposed to UVA (200 mJ/cm^2^) irradiation. Values are expressed as mean ± SD (*n* = 3). The symbol # indicates that it is significantly different from the cells without UVA exposure and HSYA pretreatment. The symbol * indicates that it is significantly different from the cells exposed to UVA irradiation and without HSYA pretreatment at *p* < 0.05, determined by one-way ANOVA followed by Tukey’s post hoc test.

**Figure 4 ijms-25-07573-f004:**
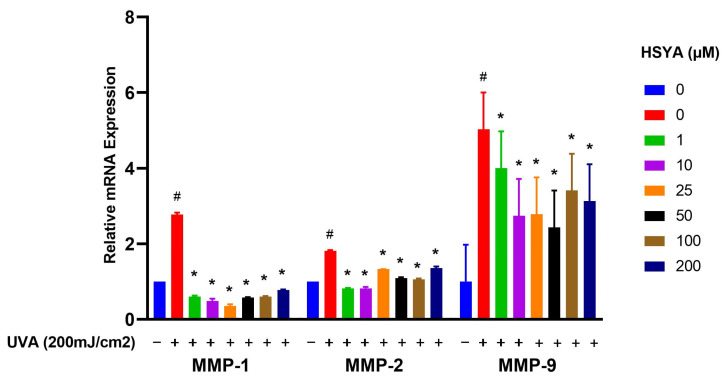
Semi-quantitative RT-PCR analysis of matrix metalloproteases MMP-1, MMP-2, and MMP-9 mRNA levels in HaCaT cells after being treated at different concentrations with HSYA exposed to UVA (200 mJ/cm^2^) irradiation. The graph represents the mRNA levels of MMP-1, MMP-2, and MMP-9 relative to the control gene Glyceraldehyde-3-phosphate dehydrogenase (GAPDH). Values are expressed as mean ± SD (*n* = 3). The symbol # indicates that it is significantly different from the cells without UVA exposure and HSYA pretreatment. The symbol * indicates that it is significantly different from the cells exposed to UVA irradiation and without HSYA pretreatment at *p* < 0.05, determined by one-way ANOVA followed by Tukey’s post hoc test.

**Figure 5 ijms-25-07573-f005:**
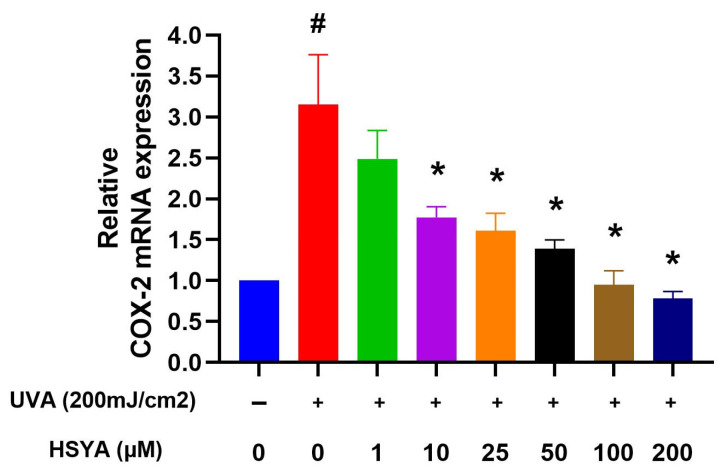
Semi-quantitative RT-PCR analysis of COX-2 mRNA levels in HaCaT cells after being treated at different concentrations with HSYA exposed to UVA (200 mJ/cm^2^) irradiation. The graph represents the mRNA levels for COX-2 relative to the control gene GAPDH. Values are expressed as mean ± SD (*n* = 3). The symbol # indicates that it is significantly different from the cells without UVA exposure and HSYA pretreatment. The symbol * indicates that it is significantly different from the cells exposed to UVA irradiation and without HSYA pretreatment at *p* < 0.05, determined by one-way ANOVA followed by Tukey’s post hoc test.

## Data Availability

The original contributions presented in the study are included in the article; further inquiries can be directed to the corresponding authors.

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
