# Peer review of "Effects of Hydroxysafflor Yellow A (HSYA) on UVA-Induced Damage in HaCaT Keratinocytes"

_ijms, 2024, doi:10.3390/ijms25147573_

Round 1
Reviewer 1 Report
Comments and Suggestions for Authors
i read with great interest about the antioxidant abilities of hydroxysafflor yellow A on keratinocytes.
some suggestions:
intro:A) GIVE A TERMINOLOGY OF OXIDATIVE STRESS
B) REPORT SKIN DISEASES DUE TO OXIDATIVE STRESS FROM uv MAINLY SKIN CANCER ( OXIDATIVE DAMAGE MOLECULES RISE , ANTIOXIDANT ENZYMES AND NON-ENZYMATIC ANTIOXIDANTS FALL (https://doi.org/10.3390/stresses3040054))
C) ALSO A BRIEF REPORT ABOUT OXIDATIVE STRESS MARKERS SUCH AS MDA FOR LIPID PEROXIDATION ETC. ETC,
D) NICE PRESENATION -MAYBE A BRIEF PARAGRAPH PRESENTING THE SUMMARY OF OXIDATIVE MARKERS AFFECTED BY THE HERB
E) VERY GOOD DISCUSSION.. MAY A REPORT ABOUT THE ANTIOXIDANT PROPERTIES OF VITAMIN D PRODUCED BY SUNLIGHT IS NOT ADEQUATE 10.3390/antiox12051107TO COUNTER OXIDATIVE STRESS INDUCED THEREFORE FURTHER ANTIOXIDANT IS INTICATED
Author Response
Comment 1: GIVE A TERMINOLOGY OF OXIDATIVE STRESS
Response 1: Oxidative Stress: An imbalance between the production of reactive oxygen species (ROS) and the body's ability to detoxify these reactive intermediates or repair the resulting damage. We add on page 2 line 49.
Comment 2: REPORT SKIN DISEASES DUE TO OXIDATIVE STRESS FROM uv MAINLY SKIN CANCER ( OXIDATIVE DAMAGE MOLECULES RISE , ANTIOXIDANT ENZYMES AND NON-ENZYMATIC ANTIOXIDANTS FALL (https://doi.org/10.3390/stresses3040054))
Response 2: We add this reference, thanks.
Comment 3: ALSO A BRIEF REPORT ABOUT OXIDATIVE STRESS MARKERS SUCH AS MDA FOR LIPID PEROXIDATION ETC. ETC,
Response 3: We add more on page 7 line 160.
Comment 4. NICE PRESENATION -MAYBE A BRIEF PARAGRAPH PRESENTING THE SUMMARY OF OXIDATIVE MARKERS AFFECTED BY THE HERB
Response 4: In page 9-5. Conclusion~ Our study demonstrated the protective effects of HSYA in HaCaT keratinocytes against UVA-induced damage thought suppressing UVA-induced production of ROS and LPO and expressions of MMP-1, MMP-2, MMP-9, and COX-2. These suppressive abilities may have contributed to the increased cell viability after UVA irradiation.
Comment 5: VERY GOOD DISCUSSION. MAY A REPORT ABOUT THE ANTIOXIDANT PROPERTIES OF VITAMIN D PRODUCED BY SUNLIGHT IS NOT ADEQUATE 10.3390/antiox12051107TO COUNTER OXIDATIVE STRESS INDUCED THEREFORE FURTHER ANTIOXIDANT IS INTICATED
Response 5: We add this reference, thanks.
Reviewer 2 Report
Comments and Suggestions for Authors
The article titled ‘Effects of Hydroxysafflor Yellow A (HSYA) on UVA-Induced Damage in HaCaT Keratinocytes’ may be an useful contribution to the journal; however, few changes should be taken into consideration, in the benefit of the reader:
Line 37 where authors say ‘..which accelerate aging and in turn can lead to skin cancer’ It should be rephrased, as carcinogenesis is not necessarily a consequence of ageing.
Lines 40-41 the phrase ‘In recent years, natural compounds have gained considerable attention as skin protective agents’ lacks references.
Line 52 states ‘this study investigated the effects of HSYA on skin care’ –this phrase does not mirror adequately the aim of this in vitro study.
Lines 56-57 ‘Our laboratory has been established photoaging UVA-induced skin cell damage cell mode analysis photoaging indicators of technology’ should be clarified. Moreover, ‘human melanoma cell activation mode’ is also confusing to the reader and should be clarified
Lines 61 ‘We have cultured human dermal Hs68 cells which are human foreskin fibroblasts before =the paragraph lacks proper reference and is not clear.
Figure 1 legend states ‘Values are no significantly different at p < 0.05 by one-way ANOVA followed by Tukey 's Post Hoc Test’- however, there are 3 columns marked with * symbol, probably a significant result? This is major issue and needs proper clarification.
Section 2.2. HSYA Increased the HaCaT Keratinocyte Viability after UVA Exposure – it should be mentioned the need for LDH test as compared to MTT test. However, the figure 2 shows that as HYSA concentration goes higher, the values of viability get closer to that of control. This should be clarified or explained; what does the control sample consists of? Control sample was explosed to UV? These aspects are not readily understandable from the manuscript and should be clarified.
Authors are advised to define all abbreviations at first appearance in text.
Grammar and punctuation must also be carefully checked within the entire article.
Comments on the Quality of English LanguageMinor editing of English language is required.
Author Response
Comment 1: Line 37 where authors say ‘. which accelerate aging and in turn can lead to skin cancer’ It should be rephrased, as carcinogenesis is not necessarily a consequence of ageing.
Response 1: We have added the information of Line 37.
Comment 2: Lines 40-41 the phrase ‘In recent years, natural compounds have gained considerable attention as skin protective agents’ lacks references.
Response 2: We add it.
Comment 3: Line 52 states ‘this study investigated the effects of HSYA on skin care’ –this phrase does not mirror adequately the aim of this in vitro study.
Response 3: We have added the information of Line 52.
Comment 4: Lines 56-57 ‘Our laboratory has been established photoaging UVA-induced skin cell damage cell mode analysis photoaging indicators of technology’ should be clarified. Moreover, ‘human melanoma cell activation mode’ is also confusing to the reader and should be clarified
Response 4: More explain on Line 206, 4.2. Drug Treatment and UVA Irradiation. We revise it.
Comment 5: Lines 61 ‘We have cultured human dermal Hs68 cells which are human foreskin fibroblasts before =the paragraph lacks proper reference and is not clear.
Response 5: We revised it.
Comment 6: Figure 1 legend states ‘Values are no significantly different at p < 0.05 by one-way ANOVA followed by Tukey 's Post Hoc Test’- however, there are 3 columns marked with * symbol, probably a significant result? This is major issue and needs proper clarification.
Response 6: We revised it.
Comment7: Section 2.2. HSYA Increased the HaCaT Keratinocyte Viability after UVA Exposure – it should be mentioned the need for LDH test as compared to MTT test. However, the figure 2 shows that as HYSA concentration goes higher, the values of viability get closer to that of control. This should be clarified or explained; what does the control sample consists of? Control sample was exposed to UV? These aspects are not readily understandable from the manuscript and should be clarified.
Response 7: We revise it.
Comment 8: Authors are advised to define all abbreviations at first appearance in text.
Response 8: We have checked all abbreviations.
Comment 9: Grammar and punctuation must also be carefully checked within the entire article.
Response 9: We have checked grammar and punctuation.
Reviewer 3 Report
Comments and Suggestions for Authors
This is a relatively simple study about the preventive antioxidant effects of hydroxysafflor yellow A (HSYA) of keratinocytes after UVA exposure. Results are not surprising. Similar studies have been published on this issue, and this manuscript confirms the beneficial effects of this component of safflower.
Under my view, this manuscript would need substantial improvements for acceptance. Some of the points that should be addressed are as follow:
Introduction
Introduction is poor. The use of UVA should be justified. Skin cancer is more related to UVB than UVA. The study is focused to skin photoaging rather than skin cancer. This should be clarified, and the sentences related to cancer would be omitted. In particular, the extrapolation to melanoma should be avoided, as the study is carried out on keratinocyte cultures, and melanocytes are not used at all.
Hydroxysafflor yellow A (HSYA) is a C-glycosyl compound. In detail, it is 3,4,5-trihydroxycyclohexa-2,5-dien-1-one which is substituted by beta-D-glucosyl groups at positions 2 and 4, and by a p-hydroxycinnamoyl group at position 6. The structure and some reference about the antioxidant effect would be included to justify the presence of hydroxy groups and its polyphenolic nature.
Finally, about the final lines 51-52 of the introduction: According to refs. 15, 16, and many others, the HSYA effects are not so unclear. Introduction should be focused and improved for justifying the current study, and previous closely related studies cannot be omitted.
Results
Recommendation: Figures could be improved for a better comprehension, facilitating comparison and saving room.
For instance, the two panels of Figure 1 would be fused in just one, as the x-axis is identical (HSYA concentration) without and with UVA irradiation.
Figure 3: mmol MDA/mg protein seems to be too high. That level of MDA concentration is not usual. Please, check units. This point is related to methods.
Figure 4: Y-axis is not a percentage (%), It seems to be a ratio per one. Please, verify. Like Figure 1, the three panels could be fused in a single figure with three bars of different colors for MMP1, MMP2 and MMP9 mRNA. This presentation could be comparatively more informative. It will show how MMP9 response is much greater than others.
Discussion: This part should be largely improved. It should be proportional and focused to the described results.
To start with, intracellular MDA is increased from 100 to around 110 after UVA exposure (figure 3, panel B, whatever the concentration units are. This moderate increase (that it is exaggerated using a discontinuous Y-axis scale) is not enough to discuss the effects using the terms like "havoc" (line 132), catastrophic rupture (line 137) or cellular devastation (line 138) at the beginning of the discussion. Discussion cannot be exaggerated, but it should be proportional to the results. I agree that UVA is dangerous, but the effects cannot describe so strongly. Re-writing the first paragraph, according to line 147 (effects can be harmful, and HSYA mitigates such effects)would be ok, but the term suppressive (conclusion, line 263) is not ok. Suppression is too strong.
Moving to the second long paragraph at the discussion: The discussion should be mainly referred to the obtained results. Some comparison to other studies might be assumable, but the discussion is mostly focused to results obtained by others (15, 16) or the (unpublished?) results of same group (see line 160, it is written our findings...). These findings are related to SOD, GPX, CAT, ROO., TEU, DPH, Ferricyanide complexes.... and in sum several parameters that are not studied in this manuscript. Remind that at the introduction, all these studied were not mentioned (lines 51-52). In addition, there is no references about it. This is unacceptable to me. Lines like 185-187 are right, but they cannot be a small part of the manuscript. Lines 193-195 would be also deleted from the discussion.
Final part of the discussion: Justify the choice of COX-2 and not COX-1. The mention that COX-2 as a cancer promoter is also exaggerated. COX-2 is involved in many inflammatory responses that have nothing to do with cancer. Delete the link between Cox-2 and cancer.
Discussion should address whether HSYA is a ROS scavenger or alternatively is an enhancer of the Nrf2 and cellular the antioxidant response. The manuscript describes HYSA as a direct ROS scavenger (see conclusion, line 265), but the mentioned effect of antioxidant enzymes suggest that HYSA could be exert the effect on the antioxidant response.
Methods
4.5: Description of LDH assay is unclear. Why only cells in the wells containing the target cell maximum LDH release ? How are those control cells identified? Are those cells 100% in LDH release? The scale used at the y-axis of Figure 2 should be clarified.
4.6: At least a reference is needed, or alternatively, the assay description should be improved. It is supposed that the fluorescence detected correspond to intracellular oxidized DCFH, but the description about DCFH2 removal is unclear.
4.7: Again at least a ref. needed. TBA concentration and other important details are missing. Currently, the protocol of the assay is unreliable.
4.8: A table with the sequence of the used primers would be provided. Indicate human primers.
Minor corrections
Line 229: 2×105 (superscript)
Line 242: H2SO4, (subscript)
Line 253: A minus sign is missing
Comments on the Quality of English LanguageMinor editing.
Author Response
This is a relatively simple study about the preventive antioxidant effects of hydroxysafflor yellow A (HSYA) of keratinocytes after UVA exposure. Results are not surprising. Similar studies have been published on this issue, and this manuscript confirms the beneficial effects of this component of safflower.
Under my view, this manuscript would need substantial improvements for acceptance. Some of the points that should be addressed are as follow:
Introduction
Comment 1: Introduction is poor. The use of UVA should be justified. Skin cancer is more related to UVB than UVA. The study is focused to skin photoaging rather than skin cancer. This should be clarified, and the sentences related to cancer would be omitted. In particular, the extrapolation to melanoma should be avoided, as the study is carried out on keratinocyte cultures, and melanocytes are not used at all.
Hydroxysafflor yellow A (HSYA) is a C-glycosyl compound. In detail, it is 3,4,5-trihydroxycyclohexa-2,5-dien-1-one which is substituted by beta-D-glucosyl groups at positions 2 and 4, and by a p-hydroxycinnamoyl group at position 6. The structure and some reference about the antioxidant effect would be included to justify the presence of hydroxy groups and its polyphenolic nature.
Finally, about the final lines 51-52 of the introduction: According to refs. 15, 16, and many others, the HSYA effects are not so unclear. Introduction should be focused and improved for justifying the current study, and previous closely related studies cannot be omitted.
Response 1: We have added the information and revise it.
The structure of HSYA.
Comment 2: Results
Recommendation: Figures could be improved for a better comprehension, facilitating comparison and saving room.
For instance, the two panels of Figure 1 would be fused in just one, as the x-axis is identical (HSYA concentration) without and with UVA irradiation.
Figure 3: mmol MDA/mg protein seems to be too high. That level of MDA concentration is not usual. Please, check units. This point is related to methods.
Figure 4: Y-axis is not a percentage (%), It seems to be a ratio per one. Please, verify. Like Figure 1, the three panels could be fused in a single figure with three bars of different colors for MMP1, MMP2 and MMP9 mRNA. This presentation could be comparatively more informative. It will show how MMP9 response is much greater than others.
Response 2: We check and revise Figures.
Comment 3: Discussion: This part should be largely improved. It should be proportional and focused to the described results.
To start with, intracellular MDA is increased from 100 to around 110 after UVA exposure (figure 3, panel B, whatever the concentration units are. This moderate increase (that it is exaggerated using a discontinuous Y-axis scale) is not enough to discuss the effects using the terms like "havoc" (line 132), catastrophic rupture (line 137) or cellular devastation (line 138) at the beginning of the discussion. Discussion cannot be exaggerated, but it should be proportional to the results. I agree that UVA is dangerous, but the effects cannot describe so strongly. Re-writing the first paragraph, according to line 147 (effects can be harmful, and HSYA mitigates such effects)would be ok, but the term suppressive (conclusion, line 263) is not ok. Suppression is too strong.
Moving to the second long paragraph at the discussion: The discussion should be mainly referred to the obtained results. Some comparison to other studies might be assumable, but the discussion is mostly focused to results obtained by others (15, 16) or the (unpublished?) results of same group (see line 160, it is written our findings...). These findings are related to SOD, GPX, CAT, ROO., TEU, DPH, Ferricyanide complexes.... and in sum several parameters that are not studied in this manuscript. Remind that at the introduction, all these studied were not mentioned (lines 51-52). In addition, there is no references about it. This is unacceptable to me. Lines like 185-187 are right, but they cannot be a small part of the manuscript. Lines 193-195 would be also deleted from the discussion.
Final part of the discussion: Justify the choice of COX-2 and not COX-1. The mention that COX-2 as a cancer promoter is also exaggerated. COX-2 is involved in many inflammatory responses that have nothing to do with cancer. Delete the link between Cox-2 and cancer.
Discussion should address whether HSYA is a ROS scavenger or alternatively is an enhancer of the Nrf2 and cellular the antioxidant response. The manuscript describes HYSA as a direct ROS scavenger (see conclusion, line 265), but the mentioned effect of antioxidant enzymes suggest that HYSA could be exert the effect on the antioxidant response.
Response 3: We revise the discussion section.
Comment 4: Methods
4.5: Description of LDH assay is unclear. Why only cells in the wells containing the target cell maximum LDH release ? How are those control cells identified? Are those cells 100% in LDH release? The scale used at the y-axis of Figure 2 should be clarified.
4.6: At least a reference is needed, or alternatively, the assay description should be improved. It is supposed that the fluorescence detected correspond to intracellular oxidized DCFH, but the description about DCFH2 removal is unclear.
4.7: Again at least a ref. needed. TBA concentration and other important details are missing. Currently, the protocol of the assay is unreliable.
4.8: A table with the sequence of the used primers would be provided. Indicate human primers.
Response 4: We add more information on methods section.
Comment 5: Minor corrections
Line 229: 2×105 (superscript)
Line 242: H2SO4, (subscript)
Line 253: A minus sign is missing
Response 5: We revise it.
Round 2
Reviewer 1 Report
Comments and Suggestions for Authors
the manuscript was significantly improved! Well done!
Author Response
Thank you very much.
Reviewer 2 Report
Comments and Suggestions for Authors
Most of the comments have not been addressed by the authors, at least in the version "peer-review-v2" that has been uploaded in the system. Therefore, I mentain my recommendation for revision.
Comments on the Quality of English Languageminor
Author Response
Comment 1: Line 37 where authors say ‘. which accelerate aging and in turn can lead to skin cancer’ It should be rephrased, as carcinogenesis is not necessarily a consequence of ageing.
Response 1: Thank you for your comments. We have added the information in Line 38.
Comment 2: Lines 40-41 the phrase ‘In recent years, natural compounds have gained considerable attention as skin protective agents’ lacks references.
Response 2: Thank you for your comments. We have added the information in Line 42.
Comment 3: Line 52 states ‘this study investigated the effects of HSYA on skin care’ –this phrase does not mirror adequately the aim of this in vitro study.
Response 3: We have added the information in Line 52.
Comment 4: Lines 56-57 ‘Our laboratory has been established photoaging UVA-induced skin cell damage cell mode analysis photoaging indicators of technology’ should be clarified. Moreover, ‘human melanoma cell activation mode’ is also confusing to the reader and should be clarified
Response 4: Thank you for your comments. We have added the information in Line 247-256, 4.2. Drug Treatment and UVA Irradiation.
Comment 5: Lines 61 ‘We have cultured human dermal Hs68 cells which are human foreskin fibroblasts before =the paragraph lacks proper reference and is not clear.
Response 5: We revised it.
Comment 6: Figure 1 legend states ‘Values are no significantly different at p < 0.05 by one-way ANOVA followed by Tukey 's Post Hoc Test’- however, there are 3 columns marked with * symbol, probably a significant result? This is major issue and needs proper clarification.
Response 6: Thank you for your comments. We have revised it in Line 82-85.
Comment 7: Section 2.2. HSYA Increased the HaCaT Keratinocyte Viability after UVA Exposure – it should be mentioned the need for LDH test as compared to MTT test. However, the figure 2 shows that as HYSA concentration goes higher, the values of viability get closer to that of control. This should be clarified or explained; what does the control sample consists of? Control sample was exposed to UV? These aspects are not readily understandable from the manuscript and should be clarified.
Response 7: Thank you for your comments. We have added the information in line 272-296.
Comment 8: Authors are advised to define all abbreviations at first appearance in text.
Response 8: Thank you for your comments. We have checked all abbreviations.
Comment 9: Grammar and punctuation must also be carefully checked within the entire article.
Response 9: Thank you for your comments. We have checked grammar and punctuation.
Reviewer 3 Report
Comments and Suggestions for Authors
First of all, the paper have been improved and my concerns have been mostly addressed, although the answers at the reply letter are brief. I think that you did not understand properly my suggestion concerning Figures, but anyway the new format is clearer. In summary, the amended manuscript is focused and the dicussion in proportional to the magnitude of the observed protective effects.
Author Response
We revise the discussion section. Thank you for your comments.
Round 3
Reviewer 2 Report
Comments and Suggestions for Authors
The manuscript has been impoved. It could be published in present form.
Comments on the Quality of English LanguageMinor